# Effects of a community-driven water, sanitation, and hygiene intervention on diarrhea, child growth, and local institutions: A cluster-randomized controlled trial in rural Democratic Republic of Congo

**John P. Quattrochi**[1]*, **Kevin Croke**[2], **Caleb Dohou**[3], **Luca Stanus Ghib**[4], **Yannick Lokaya**[3], **Aidan Coville**[4‡], **Eric Mvukiyehe**[5‡]

1 Department of Global Health, School of Health, Georgetown University, Washington, DC, United States of America, 2 Department of Global Health & Population, Harvard TH Chan School of Public Health, Boston, Massachusetts, United States of America, 3 World Bank Country Office, Kinshasa, Democratic Republic of Congo, 4 Development Impact (DIME) Department, World Bank, Washington, DC, United States of America, 5 Department of Political Science, Duke University, Durham, North Carolina, United States of America

‡ These authors are joint senior authors on this work.
* john.quattrochi@georgetown.edu

## Abstract

### Background

Diarrhea and growth faltering in early childhood reduce survival and impair neurodevelopment. We assessed whether a national program combining (i) funds for latrine and water upgrades; (ii) institutional strengthening; and (iii) behavior change campaigns reduced diarrhea and stunting, and strengthened local institutions.

### Methods and Findings

We collaborated with program implementers to conduct a cluster-randomized controlled trial in four provinces of the Democratic Republic of Congo (DRC). Three hundred thirty-two rural villages were grouped into 121 clusters to minimize geographic spillovers. Between 15 March and 30 June 2018, we randomly assigned, after stratifying by province and cluster size, 50 intervention and 71 control clusters. Masking of participants and interviewers was not possible. Primary outcomes were length-for-age Z-score among children under 5 years of age, caregiver-reported diarrhea in last 7 days among children under 5 years of age, and an index of community WASH institutions. The primary analysis was on an intention-to-treat basis, using a binary variable indicating whether the participant was in an intervention or control cluster. Three thousand two hundred eighty-three households were interviewed between November 2022 and April 2023, median 3.6 years post-intervention. The intervention had no effect on diarrhea (adjusted mean difference −0.01 [95% −0.05 to 0.03]). Diarrhea prevalence was high overall, at 38% in the treatment group and 42% in the control group. The intervention had no effect on length-for-age Z-scores

**Data availability statement:** Individual-level, de-identified data from this study and code to reproduce all results are publicly available in the World Bank micro-data catalogue here: https://doi.org/10.60572/5dww-3171.

**Funding:** This study was funded by the UK Foreign, Commonwealth, and Development Office (FCDO) (https://www.gov.uk/government/organisations/foreign-commonwealth-development-office), via Amendment No. 3 to the Supplemental Arrangement with the World Bank regarding Multi-Donor Trust Fund for Impact Evaluation to Development Impact (TF072617, parallel to TF072161). AC, KC, and EM were staff at Development Impact at that time. The Healthy Villages & Schools program was a DRC Government national program funded by UK's FCDO and implemented with UNICEF's support. Additional funding support is in part from Georgetown University Medical Center. The funders and implementing partners provided inputs at the design stage to ensure the study addressed policy and program priorities of importance to them. The funders had no role in data collection and analysis, decision to publish, or preparation of the manuscript.

**Competing interests:** The authors have declared that no competing interests exist.

**Abbreviations:** AEA, American Economics Association; DRC, Democratic Republic of Congo; ICC, intra-cluster coefficient; IHfRA, Innovative Hub for Research in Africa; MPN, Most Probable Number; WASH, water, sanitation, and hygiene.

in children (adjusted mean difference –0.01 [95% CI –0.15 to 0.12]). In the control group, the mean length-for-age Z-score was –2.18 (1.60 SD). Villages in the intervention group had a 0.40 higher score on the WASH institutions index (95% CI 0.16–0.65). The percentage of villages in the intervention group with an active water, sanitation, and hygiene (or just water) committee was 21 pp higher than the control group. Households in the intervention group were 24 pp (95% CI 12–36) more likely to report using an improved water source, 18 pp (95% CI 10–25) more likely to report using an improved sanitation facility, and reported more positive perceptions of water governance (adjusted difference 0.19 SD [95% CI 0.04–0.34]). The trial had several limitations, including incomplete (86%) adherence in the implementation group, the absence of baseline measures, and the reliance on self-reported outcomes for some measures.

## Conclusions

The DRC's national rural WASH program increased access to improved water and sanitation infrastructure, and created new WASH institutions, all of which persisted for at least 3.6 years. However, these effects were not sufficient to reduce diarrhea or growth faltering.

## Trial registration

The Pan African Clinical Trials Registry PACTR202102616421588 (https://pactr.samrc.ac.za/TrialDisplay.aspx?TrialID=14670).

The American Economics Association RCT registry AEARCTR-0004648 (https://www.socialscienceregistry.org/trials/4648).

## Author summary
### Why was this study done?

- 1.4 million deaths and 74 million disability-adjusted life years lost per year are attributable to unsafe water, sanitation, and hygiene (WASH), primarily due to diarrheal disease and child growth faltering.

- Of the many possible interventions to improve WASH, relatively few have been tested in randomized trials, particularly in conflict-affected settings, and most existing studies stop follow-up after 1 or 2 years.

- The Democratic Republic of Congo's national Healthy Villages and Schools program that we evaluated is a relatively rare combination of support for new institutions, funding for new or improved infrastructure, and a behavior change campaign, all within a locally-led process of targeting and implementation.

### What did the researchers do and find?

- We conducted a cluster-randomized trial of the Healthy Villages and Schools program, enrolling 332 villages in 121 clusters. Diarrhea prevalence, child length-for-age, WASH institutions, and related outcomes were measured 3.6 years after the program ended.

- The program did not reduce diarrhea, nor did it increase child length-for-age. The most likely explanation for these null results is that water quality did not meaningfully

improve, despite improvements to WASH institutions and both water and sanitation infrastructure.

## What do these findings mean?

- Even in conflict-affected settings, government-led programs can generate sustainable increases in WASH infrastructure and institutions.

- However, improved water sources may not be sufficient to improve water quality.

- Reducing diarrhea and increasing child length-for-age requires a more effective intervention than the one studied here.

## Introduction

The most recent estimates of the global burden of morbidity and mortality attributable to unsafe water, sanitation, and hygiene (WASH) are that 1.4 million deaths and 74 million disability-adjusted life years lost could have been prevented in 2019 [1]. People living with unsafe WASH have higher exposure to fecal-oral pathogens, resulting in enteric dysfunction, diarrheal illnesses, and, in children, growth faltering. Growth faltering, in turn, has long-term negative impacts on health, cognition, and human capital [2,3]. In 2020, 2.0 billion people did not have access to safely managed drinking water services, 3.6 billion did not have access to safely managed sanitation services, and 2.3 billion did not have access to handwashing facilities with soap and water at home [4,5]. While access has been increasing, progress will need to accelerate by 3-to-6-fold to meet the Sustainable Development Goals for 2030 [6]. The challenge of increasing access is particularly acute for people living in or near armed conflict—one in six people worldwide—both through the direct effects of conflict and because violence and insecurity impede collective action to provide public goods like WASH [7,8]. In the Democratic Republic of Congo (DRC), as of 2020, 48 million people still lacked basic drinking water services, 11 million people still practiced open defecation, and 72 million people still lacked basic hygiene services [9].

To increase access to safe WASH, governments and donors have increasingly turned to community-led approaches. While WASH experts called for greater community participation for over 30 years [10], the sector did not fully embrace this approach until the late 1990s. Beginning with community-led total sanitation in Bangladesh in 1999, community-led WASH programs have been implemented in at least 60 countries, and 15 countries have incorporated them into national policy [11,12].

Despite this broad adoption, the health effects of community-led WASH interventions—and of many WASH interventions in general—remain poorly understood. The accumulation of evidence has accelerated in recent years, but the length of the causal chain from the intervention to the outcome, and the vast design space for WASH interventions (which can incorporate behavior change campaigns, infrastructure, institutions, and/or new technologies), means that many fundamental questions remain unanswered. A meta-analysis of 13 randomized WASH trials found no effect on child length-for-age, but a meta-analysis of 124 WASH studies (randomized and observational) found a protective effect against diarrhea [11,13–24]. There is a great deal of heterogeneity in effect size across studies, likely due to variation in intervention components and intensity, and to the influence of contextual factors such as baseline exposure to fecal matter.

To our knowledge, this is the first trial of an intervention that combines the creation of new institutions, funding for new or improved infrastructure, and a behavior change campaign, all within a locally-led process of targeting and implementation. We study this complex intervention as it is implemented at scale, providing a realistic estimate of effectiveness for

policymakers in similar contexts. Our follow-up period is unusually long (3.6 years), enabling us to address the question of sustainability. We also provide evidence in a conflict-affected setting, where WASH interventions have rarely been evaluated using experimental designs.

The intervention is the 'healthy villages' component of the DRC's Healthy Villages and Schools program, co-led by the Ministries of Public Health, and of Primary, Secondary, and Professional Education, with support from the United Nations Children's Fund (UNICEF). Since 2008, nearly 9 million people in almost 11,000 villages have been reached with WASH services through the program. Healthy Villages and Schools was the largest WASH program implemented by UNICEF globally and comprised 90% of total external funding committed to rural WASH in the DRC from 2005 to 2020 [25]. Our goal was to estimate the effect of Healthy Villages and Schools on diarrhea prevalence, child length-for-age, and WASH institutions.

## Methods

### Study design and participants

The intervention of interest is a DR Congo government-run program that began several years before our study. For our study, the government agreed to randomly assign the next phase of the program (i.e., the next group of villages to receive the intervention). In 2018, we randomly assigned groups of villages to intervention or control (details below). Since we did not collect any data prior to randomization, we did not yet register the study. We collected the first round of data in late 2019, about 5 months after the intervention was implemented (implementation took about one year in each group of villages) [26].

We pre-registered the analysis of that first round of data collection (5 month follow-up) before any of the authors saw the data, in the American Economics Association (AEA) Registry (AEARCTR-0004648) (https://www.socialscienceregistry.org/trials/4648).

In February 2021, we registered plans for additional data collection in the Pan-African Clinical Trials Registry (PACTR202102616421588; https://pactr.samrc.ac.za/TrialDisplay.aspx?TrialID=14670). In April 2023, before the principle investigators had seen any data for the 3.6-year follow-up (i.e., the data for the current manuscript), we updated the registration with our primary and secondary outcomes. In April 2023, we also posted our pre-analysis plan for the 3.6 year follow-up on the AEA Registry (see "Three Year Follow up Pre-Analysis Plan" at https://www.socialscienceregistry.org/trials/4648). All planned studies from this project are now registered and any future work will be registered prospectively. This study is reported as per CONSORT guidelines (S1 Text).

We worked with intervention implementers to design a cluster-randomized trial in rural villages in five DRC provinces: Kongo Central, Kasai, Kasai Central, North Kivu, and South Kivu. The implementers identified 403 candidate villages in which the intervention could be launched during the study period, based on the established criteria for the intervention: that the village was located in a secure and accessible Health Area that was not already served by the WASH Consortium, the Health Area staff were dynamic and interested in participating, and there was a problem of diarrhea, cholera, and/or malnutrition. Thirty-four of these villages already had program activities in process before research activities began, leaving 369 eligible villages.

To avoid spillover effects from treatment villages to control villages, we grouped those villages into clusters. We considered any villages within 2.5 km of each other (using Euclidean distance between village centroids) to be part of the same cluster. Therefore, all clusters have at least 2.5 km between them. We relax this rule in South Kivu, where density is greater, and use a minimum distance of 1 km. In total, this resulted in 124 clusters. North Kivu had only

three clusters (covering 30 villages); as a result, it was not logistically feasible to include these villages in the trial. That left 121 clusters (339 villages) in four provinces.

Each village in intervention clusters received the intervention, as described above. Villages in control clusters did not receive any intervention. Data collection procedures were identical in the two groups.

The study protocol was approved by the Institutional Ethics Committee of the *Institut Superieur des Techniques Médicales de Bukavu* (DRC) (#001/2019 and #008/2022) and by Solutions IRB (USA) (#2019/10/20). With direction from the study investigators, Innovative Hub for Research in Africa (IHfRA) was responsible for data collection. No study data was collected by intervention implementers.

All residents of the targeted villages who had lived in the village for at least 4 years (i.e., moved to the village prior to the intervention) were eligible to participate in the study. Two groups of households were interviewed: households that were interviewed at 5 months post-intervention (4 per village) and households that had not been previously interviewed (6 per village). Both groups were randomly selected (at their respective entries into the study) as follows: from the center of the village, interviewers went in opposite directions to the $n$th household, where $n$ was a randomly selected number between 1 and 20. We interviewed the head woman of the household. We also asked to measure the height and weight of the youngest child aged 2–5 years, or, if none, the oldest child aged 0–2 years. Adult participants provided informed consent verbally, which was recorded electronically.

## Randomization and masking

A total of 339 villages in 121 clusters were eligible for randomization. In all provinces except Kasai Central, each cluster was given equal probability of being selected for treatment or control. In Kasai Central, due to budget constraints, we increased the probability of being selected into the control group to 75%, to reflect the fact that only 16 out of 81 villages could receive the intervention. Thus, the allocation probability for the intervention group was 25%.

We block randomized by province and number-of-villages-per-cluster (12 blocks total; see S1 Table). Since randomization was based on clusters, but the implementing organization's operational targets were based on villages, it was not possible to force the randomization to select the exact number of villages targeted without introducing potential bias. Instead, we compared the number of target villages per province to the number of treatment villages selected after randomization. In cases where the number of intervention villages was larger than the operational target, we randomly dropped an equal number of intervention villages from the largest control and intervention clusters until operational targets were met. We dropped two villages in Kongo Central and four villages in Kasai. We also dropped one control village in Kasai due to a coding error. This left 146 villages in 50 clusters in the intervention group, and 186 villages in 71 clusters in the control group. S1 Table shows how intervention and control villages and clusters are distributed across provinces. Randomization was done by the research team in Stata.

Due to the participatory and visible nature of the intervention, neither participants nor data collectors were masked to treatment status. However, data collectors did not participate in intervention implementation and were employed by a separate, independent organization.

## Procedures

The intervention, "Healthy Villages and Schools", was developed by the DRC government and UNICEF. We focused on the village rather than the school component. This program mobilizes communities to become a "Healthy Village" with 3–6 months of support from government health officials and local NGOs, including approximately $2,000 of financing for new

or improved water infrastructure, $2,000 for new or improved sanitation infrastructure, and $3,000 for personnel costs, per village. The mean village size in the intervention group was 456 people (median 400; IQR 502). The seven norms to become a Healthy Village are:

1. There is a dynamic village WASH committee.

2. At least 80% of the population has access to safe drinking water.

3. At least 80% of households use a hygienic latrine.

4. At least 80% of households dispose of their household waste hygienically.

5. At least 60% of the population washes their hands before eating and after going to the latrine.

6. At least 70% of the population is aware of fecal-oral disease transmission and how to prevent this.

7. The village is cleaned at least once a month.

The program is implemented in nine steps (Table 1) [27].

The IHfRA data collection team used electronic tablets and transmitted data to a cloud-based server, allowing the research team to conduct quality control measures in real-time, checking for consistency and errors. Additionally, IHfRA randomly selected 15% of villages for a second round of interviews, by different interviewers, with a shorter questionnaire, to check consistency across key variables. Separately, children from two households per village had their height and weight re-measured by an IHfRA supervisor, as a quality check.

## Outcomes

Primary outcomes were caregiver-reported diarrhea in the last seven days among all children who were under 5 years old at the time of the survey, length-for-age Z-score for a randomly selected child in each household, and a WASH governance index. If the household had one child between age 2 and 5, we measured the length and weight of that child. If the household had more than one child between age 2 and 5, we randomly selected one child. If the household had no children between age 2 and 5, but one or more children between age 0 and 2, we measured length and weight of the eldest child between age 0 and 2. Salter scale (Model 235 6S) and wall-mounted measuring rods (portable baby/child length/height measuring system) were used.

The WASH governance index combined questions about the presence of a water committee, frequency of committee meetings, WASH expenses (excluding maintenance), presence of a maintenance plan, whether the committee tracks health conditions in the community, and whether it tracks hygiene and sanitation.

Secondary outcomes were access to improved water and sanitation facilities, water quality at water points and in homes, hygiene knowledge and behaviors, observed handwashing, perceptions of WASH governance, children's school absenteeism, child weight-for-age Z-score, and child weight-for-height Z-score.

Structured observation of handwashing was done in four households per village. We first attempted to observe the four households that were interviewed at the five-month follow-up; if any were unavailable or unwilling, we randomly selected from the six new households to replace them. A research assistant spent 2 h in each of these households, recording if handwashing occurred at critical junctures such as before preparing food (see S8 Table for full list of junctures). This took place before the interview, to minimize Hawthorne effects.

Access to improved water and sanitation was self-reported, i.e., respondents reported whether their main water source is improved or not according to the Joint Monitoring

**Table 1. The Healthy Village and Schools program's nine steps.**

| Step | Description |
|------|-------------|
| 0 | The community learns about the program and collectively decides to adopt it before submitting a formal request to the relevant Health Zone. (A Health Zone is a geographic unit of the Congolese health system that contains roughly 10 Health Areas and 100,000 residents, run by a Chief Medical Officer (CMO)). Program protocols state that the entire community should be involved in the decision to participate. |
| 1 | A statement of agreement between the community and the Health Zone is signed. |
| 2 | Health Zone officials survey 19 households on knowledge, attitudes, and practices (KAP). The community self-evaluates on eight practices, including handwashing, water use, and sanitation. |
| 3 | The community spends about 11 h over 5 days creating calendars and maps, visiting water points, classifying hygienic practices as healthy or unhealthy, discussing fecal-oral disease transmission, calculating medical costs, and assessing which individuals and organizations influence sanitation and hygiene in the community. This includes 1.5–2 h in a facilitated activity around the question, "What are the hygiene practices that we want to change in our village?" |
| 4 | The Health Zone provides training for 20 volunteers on maintenance of latrines, water supply systems, and sanitation, conflict management, and petty cash management. The community elects a village WASH committee. |
| 5 | The community spends 10 h over three days describing a community vision, analyzing the barriers to reducing diarrheal diseases, choosing improvements to drinking water, sanitation, and hygiene, and formulating an action plan. The community is asked to identify practical, low-cost solutions with a minimum of outside assistance. New infrastructure is evaluated in terms of accessibility, technical feasibility, and technical capacity. |
| 6 | The community builds new infrastructure over 90–180 days, supported by project funds. Key messages about sanitation and hygiene are discussed during sensibilization meetings or during visits to families by the WASH committee, community health workers, or other volunteers. Health Zone staff are expected to visit the community monthly during this time; Health Area staff weekly. |
| 7 | The community self-evaluates again, to measure progress since Step 2. The Health Zone conducts additional KAP surveys and hosts 3 h of meetings to assess the findings and make a plan to maintain progress. |
| 8 | The CMO spends 1 day in the community to assess whether or not the community has completed its action plan and achieved the seven norms. If they have, a certification ceremony is held. The CMO and the village WASH committee develop a Community Action Plan for Maintenance so that the changes achieved through the program can be sustained over time. |

Program standard definitions (e.g., boreholes are considered improved, while unprotected springs or surface water sources are not). Cost paid by households for water and time spent collecting water were also self-reported.

To measure water quality, we tested samples collected (i) at each of the water points used by members of each village, and (ii) at household water storage containers in six randomly selected households per village, on average. Testing was done concurrently with the household interviews. We used the Aquagenx Compartment Bag Test *E. coli* + Total Coliform Most Probable Number (MPN) Kit. This measured the MPN of fecal indicator bacteria [28].

To measure subjective performance of local WASH institutions, we used survey questions about: fairness of selection of water governance entity, perception of fair treatment, confidence in the entity's management of money, confidence in the entity's response to infrastructure breakdowns, confidence in management, and overall satisfaction. The data collection team also directly observed water point functionality. Length of breakdowns was reported via a water committee and village leader survey.

## Statistical analyses

The number of village clusters in the study was determined by the program budget and proximity of villages to one another (details above). Sample size calculations were used to

determine how many households in each village should be interviewed. Based on the primary outcome of diarrhea prevalence, with a minimum detectable effect of 8 percentage points (pp), 32% prevalence in the control group (based on the 5-month follow-up results), intracluster correlation of 0.09 (based on 5-month follow-up), and 1.3 children under 5 years per household (based on 5-month follow-up), we required 10 households per village.

We estimated intervention effects according to random assignment (intention to treat), irrespective of adherence to the intervention.

For both primary and secondary outcomes, whether binary or continuous, we fitted linear models with a binary variable indicating whether the participant was in a treatment or control cluster [29]. Because randomization was stratified by province and number-of-villages-per-cluster, we included binary variables (fixed effects) for each stratum ($n$ = 12) in the model. We also included gender and age (month) indicator variables for all child health outcomes. We clustered standard errors at the cluster (i.e., group of villages) level.

For the primary outcome of WASH institutions, and secondary outcomes consisting of multiple measures, we calculated a summary index to avoid over-rejection of the null hypothesis due to multiple inferences. We rescaled each outcome so that higher values implied better outcomes, and averaged standardized values relative to the control group. Treatment effects were estimated as the difference in the summary index between treatment and control groups, such that treatment effects are expressed in standard deviation units relative to the control group.

We pre-registered two analyses restricted by subgroup: by province (for all three primary outcomes), and by gender (for diarrhea and length-for-age). We test for interaction on the additive scale, using interaction terms in linear models.

Statistical analyses were conducted in Stata version 16.0.

## Results

Of the 1,312 respondents in 328 villages interviewed for the 5-month follow-up, 1,133 (86%) in 327 villages were re-interviewed at the 3.6-year follow-up, between November 24, 2022 and February 10, 2023 (Fig 1). We also reached two villages that were not accessible during 5-month follow-up. In 39 households with a 5-month follow-up interview, a new respondent was interviewed at 3.6 years, and 140 households (11%) were replaced between 5-month and 3.6-year follow-ups. Additionally, in each village at the 3.6-year follow-up, six never-previously-interviewed households were randomly selected, conditional on having lived in the village for at least 4 years, yielding 1,970 interviews (in four villages, only five households were reached). Thus, at 3.6 years, we interviewed a total of 3,283 households. Of those households, 75% (2,466 out of 3,283) had at least one child eligible for a caregiver's reports of diarrhea, and 72% (2,374 out of 3,283) had at least one child eligible for length and weight measurement. The primary outcome of WASH institutions was measured in 329 villages. In the intervention group, the median time since the completion of Healthy Villages Step 6 (construction of infrastructure) was 3.6 years (IQR = 3.4–3.7).

At 3.6 years, respondents in intervention and control groups were similar with regard to characteristics unlikely to be affected by the intervention, such as marital status, educational attainment, age, religion, household size, and home construction materials (Table 2).

In the intervention group, 96% of villages reported that they created a community action plan and prioritized actions to improve WASH (as instructed by the program); 86% reported that they had implemented that plan.

The intervention had no effect on diarrhea (adjusted mean difference −0.01 [95% −0.05 to 0.03]) (Table 3). Diarrhea prevalence was high overall, at 38% in the treatment group and 42% in the control group. The intra-cluster coefficient (ICC) for diarrhea in the control group was 0.05; in the intervention group, 0.07.

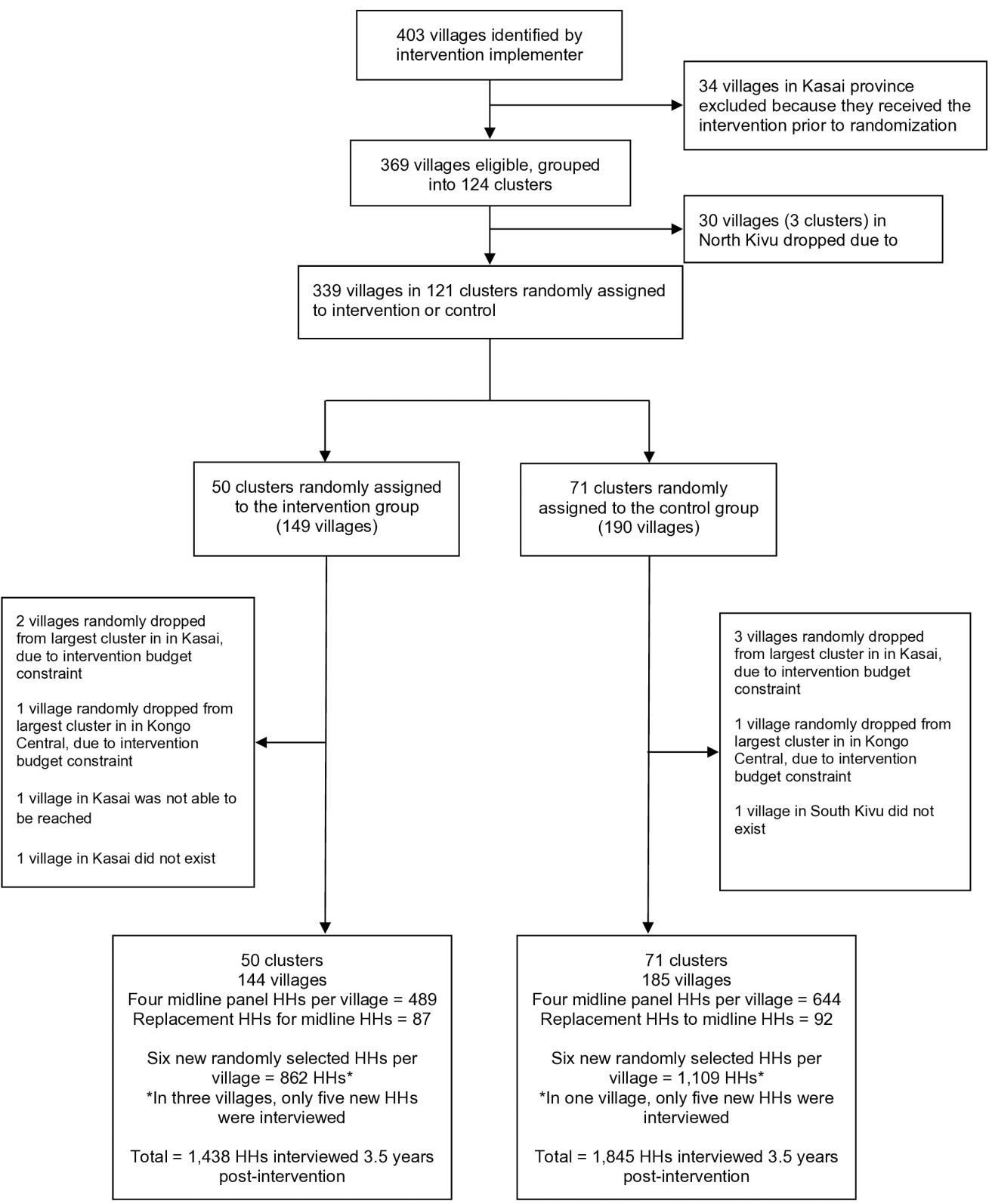

**Fig 1. Trial profile.** HH, household.

**Table 2. Household and respondent characteristics by intervention group, at 3.6-year follow-up.**

| Outcomes | Control | | | Intervention | | | | |
|---|---|---|---|---|---|---|---|---|
| | n | Mean | SD | n | Mean | SD | Adj. Diff. | p-value |
| Household has improved roof | 1843 | 0.42 | 0.49 | 1,436 | 0.47 | 0.50 | −0.01 | 0.81 |
| Household has improved wall | 1845 | 0.01 | 0.09 | 1,438 | 0.01 | 0.12 | 0.01 | 0.18 |
| Household has improved floor | 1816 | 0.04 | 0.19 | 1,430 | 0.08 | 0.27 | 0.03 | 0.11 |
| Household size | 1845 | 7.20 | 2.86 | 1,438 | 7.18 | 2.93 | 0.03 | 0.80 |
| Respondent identifies as Catholic | 1845 | 0.18 | 0.38 | 1,438 | 0.19 | 0.40 | 0.02 | 0.54 |
| Respondent identifies as Protestant | 1845 | 0.31 | 0.46 | 1,438 | 0.32 | 0.47 | −0.06 | 0.05 |
| Respondent identifies with other religion | 1845 | 0.02 | 0.15 | 1,438 | 0.03 | 0.18 | 0.01 | 0.26 |
| Respondent age | 1845 | 40 | 13.39 | 1,438 | 40 | 13.07 | 0.72 | 0.25 |
| Respondent has completed primary school | 1845 | 0.31 | 0.46 | 1,438 | 0.34 | 0.48 | −0.02 | 0.50 |
| Respondent has completed secondary school | 1845 | 0.06 | 0.23 | 1,438 | 0.07 | 0.25 | 0.00 | 0.78 |
| Respondent is married or cohabitating | 1845 | 0.83 | 0.37 | 1,438 | 0.82 | 0.38 | −0.01 | 0.49 |

Adj. diff., adjusted difference between intervention group and control group, estimated with models that include controls for randomization blocks based on province and number of villages per cluster, and standard errors clustered by cluster. There were 121 clusters in total. Improved roof = 1 if roof is finished roofing (i.e., metal, wood, calamine/cement fiber ceramic tiles, cement or roofing shingles); improved walls = 1 if walls are 'finished walls'; improved floor = 1 if floor is 'finished floor'. All variables are binary except 'HH size' and 'respondent age'; for these binary variables, the mean represents the proportion of respondents who are in the listed category.

**Table 3. Intervention effects on primary outcomes: diarrhea, length-for-age, and WASH institutions.**

| Outcomes | Control | | | | Intervention | | | | ITT | CI 95% | |
|---|---|---|---|---|---|---|---|---|---|---|---|
| | n | Prevalence/mean | SD | ICC | n | Prevalence/mean | SD | ICC | | | |
| Diarrhea prevalence | 2,310 | 42% | | 0.05 | 1,762 | 38% | | 0.07 | −0.01 | −0.05 | 0.03 |
| Length-for-age Z-score | 1,223 | −2.18 | 1.60 | 0.04 | 919 | −2.20 | 1.59 | 0.06 | −0.01 | −0.15 | 0.12 |
| WASH institutions index | 185 | 0.00 | 1.00 | 0.47 | 144 | 0.46 | 0.75 | 0.11 | 0.40 | 0.16 | 0.65 |

ITT, intention-to-treat effect estimate; ICC, intracluster correlation; HH, household. Effects are estimated with models that include controls for randomization blocks based on province and number of villages per cluster. There were 121 clusters in total. The WASH institutions index was calculated by rescaling each variable in the index (e.g., presence of WASH committee) so that higher values imply better outcomes, then standardizing relative to the control group, following Kling *and colleagues*. Effects are in standard deviation units.

The intervention had no effect on length-for-age Z-scores in children (adjusted mean difference −0.01 [95% CI −0.15 to 0.12]). In the control group, the mean length-for-age Z-score was −2.18 (1.60 SD) (Fig 2). The ICC for length-for-age Z-score in the control group was 0.04; in the intervention group, 0.06.

Villages in the intervention group had a 0.40 higher score on the WASH institutions index (95% CI 0.16–0.65). The percentage of villages in the intervention group with an active WASH (or just water) committee was 21 pp higher than the control group. The ICC for the WASH institutions index in the control group was 0.47; in the intervention group, 0.11.

Households in the intervention group were 24 pp (95% CI 12–36) more likely to report using an improved water source, 18 pp (95% CI 10–25) more likely to report using an improved sanitation facility, and reported more positive perceptions of water governance (adjusted difference 0.19 SD [95% CI 0.04–0.34]) (Table 3). The more positive perceptions of water governance were driven by higher reported satisfaction with water access (0.56 points higher (95% CI 0.27–0.85) on a 1–5 scale). Intervention group households were also 9 pp more likely to report paying for water (95% CI 0–19). Conditional on paying for water, there was no difference in the amount paid between the intervention and control groups. The intervention had no effect on time spent collecting water.

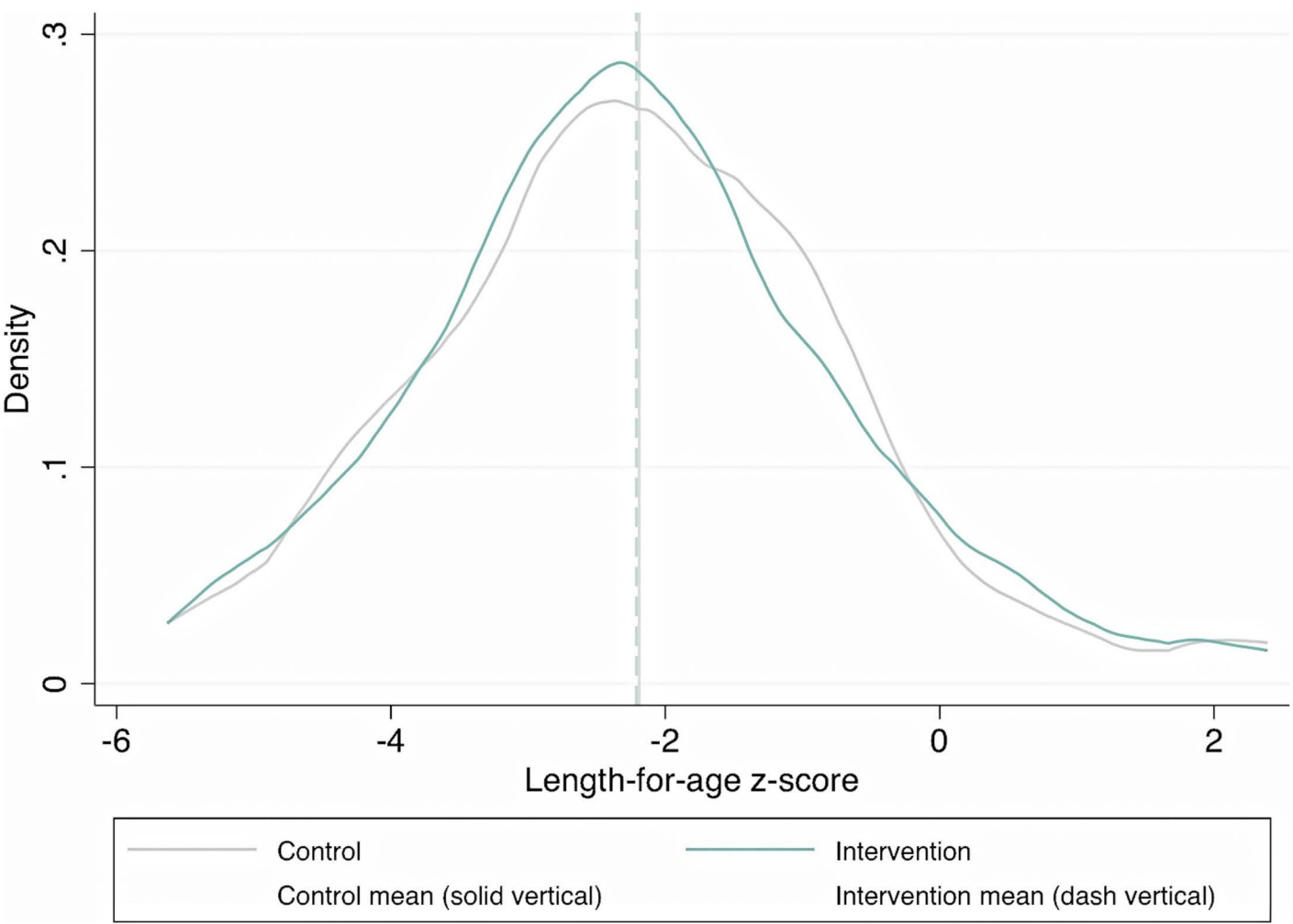

**Fig 2. Distribution of length-for-age Z-scores, intervention and control groups.** Length-for-age Z-scores for children aged 0–5 years, in the intervention group (*n* = 919) and the control group (*n* = 1223).

Intervention group water points were 11 pp less likely to be currently functional (95% CI −0.18 to −0.05); 97% of water points in control villages and 85% of water points in intervention villages were functioning.

Intervention group respondents scored 0.23 higher (95% CI 0.12–34) on the index of self-reported hygiene and behavior index. This was driven by several measures. The intervention group was 3 pp more likely to report treating their water (95% CI 1–5), and 9 pp more likely to report handwashing with soap or ash at least once in the previous day (95% CI 6–13). The intervention group also scored 0.43 points higher (95% CI 0.19–0.67) on the handwashing knowledge scale (range 0–10). However, in structured observations of handwashing behavior, there was no difference between intervention and control households in measures of any observed handwashing or handwashing with soap or ash.

Water samples from intervention village water points showed a small but statistically significant improvement in thermotolerant coliforms per 100 mL compared to control village samples (−0.17 adjusted mean difference in log10(MPN); 95% CI −0.32 to −0.02). Overall water quality was low, even from improved sources. Among unimproved sources, 86% of samples had coliform levels over 100 per 100 mL, 'very high risk' according to WHO standards [30];

among improved sources, 60% of samples had coliform levels over 100 per 100 mL. Water samples from intervention household water containers showed no difference in thermotolerant coliforms per 100 mL compared to control household samples.

The intervention had no effect on the psychological well-being index, the life satisfaction and self-esteem index, or school attendance.

In the pre-specified subgroup analysis of primary outcomes by province, we find that the intervention reduced diarrhea in one of four provinces (Kongo Central), reduced length-for-age in one province (Kasai Central), and increased the WASH institutions index in two provinces (Kasai and Kasai Central) (see S3 Table). We also used interaction terms in linear models to test for effect modification on the additive scale (S5 Table). Of the nine coefficients (three provinces, leaving out a reference, and three primary outcomes), two were statistically significant: in Kasai Central province, the results suggest that the intervention increased diarrhea prevalence relative to the treatment effect in the reference province (Kongo Central); in Kasai province, the results suggest that the intervention increased length-for-age. For both diarrhea and length-for-age, the Wald test rejects the null hypothesis that all province-by-intervention coefficients are zero at the 0.05 level. For the WASH institutions index, the Wald test fails to reject the null.

In the pre-specified subgroup analysis of diarrhea and length-for-age by child sex, we find no difference in intervention effects by sex (see S4 Table). In linear models with interaction terms for intervention-by-sex, the coefficients are not statistically significant (S7 Table).

## Discussion

We tested the effects of the national community-led rural WASH program in the DRC on child length-for-age, diarrhea, and WASH institutions. The program improved community WASH institutions, with intervention villages more likely to have a WASH committee, and for this institution to actively monitor community health conditions. Intervention villages also had greater access to improved water and sanitation infrastructure as a result of the program. However, we cannot reject the null hypothesis that the intervention did not have any effect on child length-for-age or diarrhea.

The finding of no effect on length-for-age is consistent with a recent meta-analysis of 11 WASH trials with length-for-age as a primary outcome that found an adjusted mean difference in Z-score between intervention and control of 0.00 (95% CI −0.03 to 0.04) [13]. These trial results stand in contrast to many observational studies finding that WASH protects against growth faltering [31]. This suggests that the observational results may be confounded by other household or community characteristics.

We measured fecal indicator bacteria in water sources and household water containers. We found no difference between the intervention and control group household water quality, despite the fact that intervention households were 24 pp more likely to use an improved water source. This is likely due to the fact that improved water sources in our study had low-quality water, consistent with evidence from DRC [25] and elsewhere [32]. It may also be linked to recontamination of water between collection and its ultimate use in the household. Overall, intervention water points had only slightly lower levels of fecal indicator bacteria than control water points; log10MPN/100 mL was 0.17 lower in intervention water points (95% CI −0.32 to −0.02), with a mean of 1.71 in the control group. This is consistent with a meta-analysis of five WASH trials that found only a 6% reduction in prevalence of enteropathogens in environmental samples [33].

In addition, two-thirds of our respondents reported spending at least one day per week working on an agricultural plot (modal response = 4 days). Of those who did, 95% report

open defecation while in the field, and 91% report drinking from surface water or unprotected springs. This highlights the challenge of delivering safe and comprehensive WASH services in some agricultural settings.

Our finding of no effect on diarrhea contrasts with a meta-regression (conducted as part of a systematic review) which found use of an off-site improved water source reduces the relative risk of diarrhea by 19% compared to unimproved water [14]. The same analysis found that basic sanitation without sewer connection lowered the relative risk of diarrhea by 21% relative to unimproved or limited sanitation, which also contrasts with our results, while hygiene interventions reduced the relative risk of diarrhea by 30%. However, another review found that effective handwashing promotion typically requires daily to fortnightly contact between the promoter and participant [31]. It is possible that our intervention achieved that frequency of contact during the most intensive 90–180 days of implementation, but also likely that effects would have faded out by our measurements over 3 years later.

We found no effect of the intervention on handwashing with soap or ash during structured observations of study participants by our research team. However, participants in the intervention group were 9 pp more likely to report washing with soap or ash the previous day. This underscores the limitations of self-reported data, particularly when the socially desirable outcome is likely to be known and salient.

Sustainability is a widely-recognized challenge for WASH interventions. At 5 months post-intervention, Healthy Villages and Schools increased access to improved water sources and improved sanitation facilities[26]. Notably, these improvements largely persisted to 3.6 years post-intervention, as did the improvements in WASH institutions. Yet these improvements did not result in any measurable effect on diarrhea or growth faltering. This highlights that it is crucial to measure health outcomes directly and not assume that better inputs are sufficient to yield improvements.

This study has several limitations. First, the trial had incomplete adherence: 86% of villages in the intervention group reported that they had implemented the community action plan to address WASH challenges (e.g., by building new infrastructure) by the 3.6-year follow-up. However, this is a realistic level of adherence for a government-implemented program. Indeed, given that many study villages were conflict-affected, the take-up rate was substantial. Second, we have no baseline measures. Although the randomized design means that such measures are not required for unbiased estimates of treatment effects, there are theoretical challenges; for example, if permanent migration out of study villages was affected by the intervention, then our estimates may be biased. However, at 3.6 years post-intervention we were able to re-interview 88% of the households interviewed at 5 months post-intervention, suggesting that migration was relatively rare in our study population. We also restricted households that were newly recruited at 3.6 years to those who had lived in their current residence for at least four years. Third, the outcomes that are self-reported may suffer from reactivity or social-desirability bias.

Our results reinforce calls for more ambitious attempts to improve WASH services to reduce stunting and diarrhea, such as "transformative WASH" [34]. Proponents of transformative WASH argue that some or all of the following may be necessary to produce significant health gains: high community coverage of improved sanitation facilities; living environments free from animal feces; continuous, convenient access to clean water; new approaches to behavior change; or new technologies to deliver WASH services [31]. Others go further and argue for transformative housing, with connections to water and sanitation networks [12]. These critiques are relevant in our setting, given the multiple potential sources of contamination, the low quality of even "improved sources", and the high burden of disease. Business as usual is not enough.

## Supporting information

**S1 Table. Randomization strata.**
(DOCX)

**S2 Table. Intervention effects on WASH institutions index and index sub-components.**
(DOCX)

**S3 Table. Intervention effects on all secondary outcomes, including index sub-components.**
(DOCX)

**S4 Table. Intervention effects on all primary outcomes, separately by province (pre-specified).**
(DOCX)

**S5 Table. Intervention effects on all primary outcomes, province-by-intervention interaction models.**
(DOCX)

**S6 Table. Intervention effects on diarrhea and length-for-age Z-score, separately by sex (pre-specified).**
(DOCX)

**S7 Table. Intervention effects on diarrhea and length-for-age Z-score, sex-by-intervention interaction models.**
(DOCX)

**S8 Table. Variable definitions.**
(DOCX)

**S1 Text. Consort checklist.**
(DOCX)

**S2 Text. Protocol and pre-analysis plan, 5-month follow-up.**
(DOCX)

**S3 Text. Protocol and pre-analysis plan, 3.6 year follow-up.**
(DOCX)

## Acknowledgments

The authors would like to thank the study participants, the data collection team from IHfRA, our partners in the DRC Ministries of Public Health and of Primary, Secondary, and Professional Education, at UNICEF, and at UK FCDO. The findings, interpretations, and conclusions expressed in this work do not necessarily reflect the views of The World Bank, its Board of Executive Directors, or the governments they represent.

## Author contributions

**Conceptualization:** John P. Quattrochi, Aidan Coville, Eric Mvukiyehe.

**Data curation:** Caleb Dohou, Luca Stanus Ghib.

**Formal analysis:** John P. Quattrochi, Caleb Dohou, Luca Stanus Ghib.

**Funding acquisition:** Eric Mvukiyehe.

**Methodology:** John P. Quattrochi, Kevin Croke, Aidan Coville, Eric Mvukiyehe.

**Project administration:** Caleb Dohou, Yannick Lokaya, Aidan Coville.

**Supervision:** John P. Quattrochi, Caleb Dohou, Yannick Lokaya.

**Writing – original draft:** John P. Quattrochi, Kevin Croke.

**Writing – review & editing:** John P. Quattrochi, Kevin Croke, Caleb Dohou, Luca Stanus Ghib, Yannick Lokaya, Aidan Coville, Eric Mvukiyehe.

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
