## [Editor Report · Decision Letter 0]

29 Jul 2024

Dear Dr Quattrochi, 

Thank you for submitting your manuscript entitled "Effects of a community-driven water, sanitation, and hygiene intervention on diarrhea, child growth, and local institutions: a cluster-randomized controlled trial in rural Democratic Republic of Congo" for consideration by PLOS Medicine.

Your manuscript was read with interest by the editorial staff at PLOS Medicine. However, before we can decide whether to take it forward to peer review, we would ask you to provide a response to the following: 

(1) We noted that participant recruitment began in 2018, though this study presented here was registered in 2021 from what we can tell. Were the outcomes stated in this study pre-specified at the time of patient recruitment?

(2) Could you send us the (original) study protocol please?

If we are satisfied that the paper fulfils our journal requirements after evaluation with regards to the 2 points above, we can then be in a position to advise you as to whether we would be willing to take this forward to peer review.

At this stage, we also ask that you complete your submission by providing the metadata that is required for full assessment. To this end, please login to Editorial Manager where you will find the paper in the 'Submissions Needing Revisions' folder on your homepage. Please click 'Revise Submission' from the Action Links and complete all additional questions in the submission questionnaire.

Please re-submit your manuscript within two working days, i.e. by Jul 31 2024 11:59PM. Of course, if you need more time, please do just let me know (ssunny@plos.org).

Kind regards,

Syba Sunny, MBBS, MRes, FRCPath

Associate Editor

PLOS Medicine

ssunny@plos.org

---

## [Decision Letter · Decision Letter 1]

26 Sep 2024

Dear Dr Quattrochi,

Many thanks for submitting your manuscript "Effects of a community-driven water, sanitation, and hygiene intervention on diarrhea, child growth, and local institutions: a cluster-randomized controlled trial in rural Democratic Republic of Congo" (PMEDICINE-D-24-02389R1) to PLOS Medicine. 

Firstly, please accept my apologies for the delay in reaching an outcome with your submission. There were discussions amongst the editorial staff that took some time to conclude with regards to the registration of both the work submitted to PLOS Medicine as well as the study that preceded this. We are grateful to you for having providing the necessary information and files, and for having been so open and transparent with us. Thank you also for bearing with us whilst we discussed this. We are very pleased to inform you that we are able now to progress to the next stage in the process.

The paper has been reviewed by subject experts and a statistician; their comments are included below and can also be accessed here: [LINK]

As you will see, the reviewers raised a number of concerns and points that will need addressing. We will also require an explicit description of the study registration issues in the Methods section. After discussing the paper with the editorial team and an academic editor with relevant expertise, I'm pleased to invite you to revise the paper in response to the reviewers' comments and editorial requests. We plan to send the revised paper to some or all of the original reviewers, and we cannot provide any guarantees at this stage regarding publication.

We ask that you submit your revision by Oct 24 2024 11:59PM. However, if this deadline is not feasible, please contact me by email, and we can discuss a suitable alternative.

Don't hesitate to contact me directly with any questions (ssunny@plos.org). 

Best regards, 

Syba 

Syba Sunny, MBBS, MRes, FRCPath 

Associate Editor

PLOS Medicine

ssunny@plos.org

Comments from the academic editor:

The academic editor was supportive of your work. However, she commented that the reviews were robust and agreed that the concerns raised required addressing in full. She is looking forward to reviewing a revised manuscript.

Comments from the reviewers: 

Reviewer #1: This manuscript reports on the results from a cluster-randomized trial of a national WASH program in the DRC. The results are consistent with other recent large studies of WASH interventions in terms of the limited impact on outcomes. A major strength of the study is the length of follow-up - 3.5 years. Some methodological details need clarification as described below. Overall, I don't feel the manuscript is sufficiently novel to warrant publication in PLOS Medicine.

Minor comments:

1. Introduction - what is VEA? The Healthy Villages and Schools program?

2. Methods - was distance between villages from the village center? Or village boundary?

3. Methods - the meaning of this is unclear "Six villages were randomly dropped to ensure UNICEF target numbers were met"

4. Methods - for stratified randomization, was number of villages in the cluster categorized in some way? 

5. In general, abbreviations are confusing and need to be defined at first use

6. Methods - its not clear what data collection was for the intervention program vs. for this research study and it would be helpful to clarify that. More clarity of the timing of measurement of each outcome would also be helpful

7. Methods - more details on the timing and samples collected for the water testing is needed

8. Statistical analyses - how could the sample size be determined based on results after follow-up started?

9. What was the rationale behind interviewing some individuals that had been interviewed at 5 months and some that weren't? were these considered differently in the analysis?

10. Discussion - it would be better to focus on the explanation of the findings that is most likely rather than listing all the options. Here there was no reduction in pathogen exposure which may be the most likely explanation?

11. Table 1 - the standard deviation for binary variables is not interpretable. Can you instead report the village prevalence median and interquartile range (to handle the geographic clustering)

12. Table 2- for diarrhea the "mean" is a prevalence? It is unclear how to interpret the standard deviation for diarrhea prevalence. This also applies to binary outcomes in Table 3

Reviewer #2: General comments

In this article, Quattrochi and colleagues report results from a large, cluster randomized trial of a national-level Water, Sanitation, and Handwashing (WASH) program in rural DRC, where they measured effects on child diarrhea, linear growth, and community WASH indicators. I commend the authors on the completion of this important study, which provides important (if disappointing), null results regarding the effect of the program on child health. The trial was conducted in a study population with extremely high burden of diarrhea (42% of children <5 y had diarrhea in the control group in the previous 7d) and linear growth faltering (mean z = -2.2 in the control group). Although not measured, the study communities presumably experience very high levels of enteric pathogen infection and likely could benefit from improved WASH conditions. 

I have some concerns about lack of detail in the outcome measurement methods, which I have listed below in major comments. Presuming those details can be clarified the trial did have good precision to rule out even small differences between groups that are clinically relevant (difference in diarrhea prevalence +/- 4%, difference in length-for-age z of +/- 0.14). Therefore, if measurement approaches were reasonable I would consider it an informative (null) trial. I hope my comments are constructive.

MAJOR comments

(1)

Page 14, Paragraph starting: "The study team randomized these into 50 intervention clusters (containing 146 intervention villages) and 71 control clusters (containing 187 control villages)."

These details about randomization seem out of place, as they should go in the Randomization and Masking section on the following page 15. 

Regarding the randomization, it would be helpful if you could describe the randomization with a few more details. I can gather that you stratified the randomization by province and cluster size. However, it would be helpful for you to report the following. 

* Number of strata in the randomization. Since cluster size (number of villages) is an integer count, were clusters matched exactly on size or within bands or nearest neighbor? If the latter, please specify the ranges used. Again, specific description of the number of provinces x cluster size strata would be helpful. 

* Then, please clarify if the randomization was simple randomization within strata or blocked. If blocked, were block sizes randomly permuted or fixed? Typically, blocking is recommended to preserve balance within strata so it would be helpful to understand these details (not the end of the world if it wasn't blocked). 

* for the 15 villages that were dropped due to "program constraints", was that done before the stratification and randomization, or after? If after, then it would undermine, slightly, the stratification by cluster size in those cases. Either way, it would be helpful to be clearer about these details.

(2)

Page 15, Procedures

It would be helpful if the authors could provide more detail about the dates of program implementation. The outcome measures are described as taking place 3.5 years after implementation, but it was unclear if this was from the start of activities or from the point when communities had progressed through all 8 steps described in the paper.

(3)

Page 17, Anthropometry outcomes

Please explain the rationale for a prioritizing anthropometric growth measurements among children ages 2-5 years old. Please provide detail about how the field team measured child anthropometry. Did the team follow a standard protocol? For example, did the team collect replicate measurements? Did they measure recumbent length amongst children <24 months? I noticed that the SDs of 1.6 for length-for-age (should it be called height-for-age since it includes children older than 24 months?) are quite high for a z-score. Typically, z-scores would have SD close to 1.0 in studies with very high quality anthropometric measurements. 

(4)

Page 17, Diarrhea outcome

It would be helpful if you could clarify how and when were caregivers interviewed? In their home? A single interview, or in multiple, monthly visits? Was diarrhea defined by caregivers or by symptoms? If using a symptom-based definition, were caregivers asked about each symptom separately or all together?

(5)

Page 17, Structured observation

Please explain the details of structured observation activities. E.g., Frequency? Duration? Key activities of focus? Coding? Standardization? This can be quite complex, and the description in the methods is scant.

(6)

Page 18. Water quality measurements. 

The method used to measure water quality seems appropriate, but there are few details in the methods of what bacterial indicators were measured. In the results, I infer thermotolerant coliforms? These are very non-specific to human pathogens, but a reasonable, coarse indicator of fecal pollution (though from all mammals). Please provide additional details. Additionally, and this is relatively minor, but the results appear to be reported in terms of difference on the natural log scale, but for water quality indicators such as total coliforms it is more conventional to report them on the log base 10 scale. Civil engineers will immediately interpret a "half log reduction" on base 10, and improvements will not be as clear to most readers in the field if reported on the natural log scale.

MINOR comments

(7)

Does VEA represent an acronym? I could not find any explanation for what it means, except that this was an evaluation of the "VEA program". 

(8)

Page 14

"…with distance determined 'as-the-crow-flies'"

I suggest avoiding colloquialisms. Do you mean Euclidean or great circle distance?

(9)

Page 14

"Stratified randomization ensures that the intervention and control groups will be in expectation statistically indistinguishable from each other with respect to these characteristics, likely increasing the precision of final estimates."

Gains in efficiency through stratification, though possible, are not due to improved balance within strata — gains depend on the correlation between outcomes in the two groups within strata. I suggest you simply omit this sentence, as you don't need to justify this element of the design, as there are many reasons to stratify beyond possible gains in precision.

(10)

Page 15, Participants

It seems that children also participated in the study, not only women. Perhaps that should be explained here as well. 

(11)

Page 18 Statistical Analysis

I was surprised to not see a sample size or detectable effect calculation for linear growth, a primary outcome for the trial. How could the trial have a primary outcome around which it was not designed? That seems a bit unconventional to me. 

(12)

Page 21

"Given the lack of an overall effect on diarrhea and length-for-age, we interpret the province-specific effects as statistical noise due to small sample sizes."

You could consider using less colloquial description than "statistical noise". It could be a chance finding or exploratory. This is a technical point, but the subgroup analyses were done by stratification and re-estimation of differences between groups. That is fine, but it fails to provide a formal test of effect modification on any scale (here, the additive scale is probably most appropriate). The authors could consider formal tests of interaction on the additive scale for each of their pre-specified subgroup analyses. Those are typically conducted using interaction terms in the linear regression models, and then a global test for model fit with and without the interaction term(s) (either Wald-type or likelihood-ratio-type tests). 

(13)

Page 21

"We measured enteropathogens in water sources…"

Please re-word: you measured "fecal indicator bacteria" in water sources, which is quite different from enteric pathogens. Additionally, in this paragraph you suggest that the reason you may have not found differences in household water quality was due to the poor source water quality, but another reason could have been contamination between the source and stored water — widely documented in previous research. Finally, this paragraph ends with "More intensive WASH interventions may be necessary to prevent growth faltering.", which seems completely out of place in my opinion. 

(14)

Table 1. It would be helpful to clarify in the caption or legend that these represent measures after 3.5 years of intervention, not at baseline

Reviewer #3: Introduction: I would recommend condensing and restructuring the introduction. You have a lot of good information; however, the global impact of community-based WASH interventions is is the majority of the intro and there is not any information to introduce the DRC context. It would be helpful to readers to link the global information to the country context in particular. For example, you mention "challenges of increasing access is particularly acute ... near armed conflict". This is an excellent segway into how this is pertinent to South Kivu, in particular. 

Please define VEA for readers

Typically at the end of the introduction the reader has a good grasp of what this article is aiming to describe and how this study will add to the current literature. However, I do not have have a clear grasp from the introduction what this manuscript is trying to accomplish. I would recommend more clearly stating your intent.

Methods: I would recommend restructuring the study design section. I believe you can condense it into 2 paragraphs and be more concise with the wording. For example, it was not clear until the end of the second paragraph whether North Kivu was included in the analysis or not. 

I would recommend putting the 9 steps of the VEA in a table or flow chart

Did you collect information on access to food, antibiotics taken, any lab component or any other factors that could influence Z scores?

Results: I advise to be careful stating that the intervention had zero effect on XYZ, it has been 3.5 years since the intervention. What were the results from the 5-mo post intervention and how do they compare to the current analysis? 

i would recommend removing: The intervention had no effect on the psychological well-being index, the life satisfaction & selfesteem

index, or school attendance. completely. as this is not pertinent to the specific aims/objectives of this particular manuscript. 

Discussion: I would recommend restructuring and focus on the lack of sustainability this intervention may have had. You also mention that the intervention water points were not significantly better than the control groups. This could be a huge transmission route for why the intervention appears to not have an effect. While the findings are very interesting and I believe could be a useful addition to the literature, I would be wary about claiming that the intervention had no effect on certain aspects. I recommend structuring the discussion on things the intervention succeeded may have succeeded in (i.e., more people seeking out a clean water source when available, or higher rates of hygiene) and then talk about where there are still gaps and how you would recommend future community-led interventions should focus on. 

It is also recommended that the discussion be shortened to one or two paragraphs re-iterating results, then using the literature to support key findings. At this time, the first half of the discussion reads more like a results section

Reviewer #4: The paper "Effects of a community-driven water, sanitation, and hygiene intervention on diarrhea, child growth, and local institutions: a cluster-randomized controlled trial in rural

Democratic Republic of Congo" by Quattrochi et al." analyzed outcomes through a randomized controlled trial in rural areas in the Democratic Republic of Congo whether a nationally implemented WASH program affected health outcomes. The authors used the VEA program as intervention and had prolonged follow up times in order to further assess the sustainability of the program. While the background and the execution of the study is well described, the methods section could benefit from additional adjustments for clarity.

Major comments:

Methods 

* I am a bit confused about your randomization strategy. While in the Study Design you write "We stratified randomization to ensure that the intervention and control groups are balanced", the Randomization and masking section states "Clusters were randomly allocated to intervention using a random number generator" and further in the Statistical analysis you write "Sample size was based on the primary outcome of diarrhea prevalence". Please clarify your randomization strategy coherently. Furthermore, please state why you did not consider village size in the randomization of the clusters as this can have a major effect with respect to the size of the intervention and control group.

* For the subgroup analysis, restricting the data to the specific subgroup and perform the analysis again is one method. However, to avoid sampling bias which might occur due to the data restriction to the subgroup, the model should be fit with a variable for each of the provinces (if province is the subgroup) and the respective coefficient should be examined. This way, sampling bias can be avoided and a more general claim can be made.

* In your discussion you write that "changes did not result in any effect". However, this claim is not supported by your analysis. Since you use linear models, you check for the effect size to be significantly different from 0 and H0 is your coefficient estimate is 0. However, if this is the case, you can only claim that you could not find an effect vs. there is no effect because these are two different statements with different consequences. Thus, please adjust your discussion accordingly. 

Minor comments:

General

* While to the data is provided as data.worldbank.org, more detail on where exactly the data can be found would be beneficial. Furthermore, a different link for getting access to the code should be provided upon publication.

* For better readability, if you write numbers with decimal places, please be consistent in their writing, i.e. central point '' 2·0 billion people'' vs decimal point ''1.4 million deaths''.

Abstract

* Findings: include CI to signify confidence interval when reporting diarrhea prevalence.

Introduction

* Please write out the abbreviation "VEA" as this is often used later in the manuscript.

Methods

* Thirty four should be written as a number 43, as only number until twelve are written as words in a flow text.

* The link to the implementation of the VEA program should be put as a reference and cited accordingly.

* Statistical analyses section: ''..we fit linear models..'' and "and fit the model described above" These sentences should be written in past tense as this is work you have done already.

* You mention that your random sampling, is "likely increasing the precision of final estimates." Since you are not performing computational statistics with Monte Carlo Methods, your random sampling targets towards dividing the population into two comparable groups for assessing the intervention effects. Thus, this part of the sentence should be deleted.

* In the statistical analysis section you state that you fitted linear models. I assume you fitted generalized linear mixed models, as the outcome variable does not always follow a normal distribution (diarrhea?) and you included random effects for the clusters. If this is the case, please state any other distributions than Gaussian for the response and change the wording to generalized linear model. Else please clarify your statistical approach.

* Please state the range of the The WASH governance index. Is it a number between 0 and 100, or 1-10, or 1-50 and provide a link or a short description in the appendix how this is calculated.

Results

* Table 1 would benefit from statistical tests, e.g. t-test, for differences in the groups with respect to the characteristics, as this would underpin your claim statistically.

* "mean length-for-age Z score was -2.18 (SD 1.60)", correct to "(1.60 SD)" to be consistent with your writing.

* For better readability state in the Methods section that the ITT (intention-to-treat) estimate corresponds to the parameter estimate of the "binary variable indicating whether the participant was in the intervention or control cluster." In this case, this number states an effect measure and is not the mean difference, as you write in the Results section. "The intervention had no effect on diarrhea (mean difference -0.01 [95% -0.05- 0.03])", in particular when diarrhea is not normal distributed. Please correct also in the abstract.

* Correct "Villages in the intervention group had a 0.40 SD higher score" and "Intervention group respondents scored 0.23 SD" to "Villages in the intervention group had a 0.40 higher score" and "Intervention group respondents scored 0.23"

* Correct "(-0.39 difference in ln(MPN); 95% CI -0.73- -0.04) ." to "(-0.39 difference in ln(MPN); 95% CI -0.73 - -0.04).".

Discussion

* The discussion section should completely be written in past tense.

Reviewer #5: Overall

This is an important long term follow-up of a pragmatic implementation of a government's evidence based intervention.

The conflict setting is indeed important and a big achievement for the researchers to have completed this project.

There is a major issue with studies of complex interventions and non-drug trials. There is an important move away from primary and secondary outcomes to causal pathway analysis and not seeing p-values as be all and end all.

In my view, there is too much emphasis in this complex evaluation on p-values, than on using the range of distal and proximal outcomes that exist in a causal pathway analysis that may better account for changes observed.

Having said that, the fundamental message that is striking is that the quality of water is not good, functionality of some infrastructures were not better in intervention, and knowledge did not transfer to change of behaviour. Even the infrastructures that remained, were minimal in their differences (arguably not clinically significant). All these likely to show in individual tests or a causal pathway analysis (if was done) that health outcomes are unlikely to be present.

Several questions could be better addressed in this manuscript which could allow the reader to get more out of the evidence.

Intervention

What were the average size of communities/villages as the unit which implemented for a group of people at village or neighbourhood or what level?

How many families per volunteer and were the volunteers still active or the committees still active - it seems these were assessed but reporting was not clear on the outcome of these

What were the infrastrucures and what funds were available for these after the first year? It seems the water points n the control communities are just as many - this will cause no difference between arms for the outcomes measured but there is little discussion of this.

What follow-up by the government or NGOs or anyone made for the sake of these projects in 3 years irrespective of water points, in terms of quality of water, behaviour change and community monitoring of each other etc ? (Any and best activities or training, if left without follow-up and encouragement, tend to become less effective, even a well trained doctor without input from colleagues and equipment and functioning clinic will wither and become ineffective) It is not clear what has been done in 3 years to support these communities - if nothing, then the findings are not surprising - this is one of the tragedies of development for many such projects.

Assessment

1. what type of obsrvaiton was actually conducted that did not have reactivity biases - did the consent or training of field-assessors allow for masking of assessment intentions for the households.

2. Were the field-assessors blind to the intervention or not?

Outcomes

Diarrhoea was very high - it is not possible that 40% of children had diarrhoea in a week even in a conflict zone at high diarrhoea season.

It is very well established that reported diarrhoeas are very poor measures of diarrhoea and especially if definition is not well explained - these figures are very odd, even if the overall results of no difference between arms is understandable.

Discussion and conclusion -

Although it is clear that these settings are very complex and other risks affect outcomes such as open defecation on going to the field for farming, there are many other things that affect outcomes and make the situation complex - these include open defaecation on travelling between villages, or food safety/hygiene, children playing on dirt and geophagy etc all have additional risks, but it is important to note that this is an RCT and so these factors should cancel out.

There is little discussion on any contamination to other villages in the control arm and if this was not possible due to infrastructure resources, then what about the behaviour related issues. For example it seems spending was same between 2 arms too as was time taken to fetch water and availability of water points.

---

* One of our most important requests is to do with the study registration. We thank the authors for their openness and transparency with this. We kindly ask that, in the Methods section of your revised manuscript, you describe the particulars surrounding the study registration for both studies (including the timing of registration in relation to patient recruitment and explain the specific reasons for failing to register the work prior to patient enrolment). We also ask that you write a sentence to confirm that all studies are now registered and that any future work will be registered prospectively

* Please upload any figures associated with your paper as individual TIF or EPS files with 300dpi resolution at resubmission; please read our figure guidelines for more information on our requirements: http://journals.plos.org/plosmedicine/s/figures. While revising your submission, please upload your figure files to the PACE digital diagnostic tool, https://pacev2.apexcovantage.com/. PACE helps ensure that figures meet PLOS requirements. To use PACE, you must first register as a user. Then, login and navigate to the UPLOAD tab, where you will find detailed instructions on how to use the tool. If you encounter any issues or have any questions when using PACE, please email us at PLOSMedicine@plos.org.

* Thank you for completing a CONSORT checklist. Could you kindly revise the checklist, using section and paragraph numbers, rather than page numbers, please? (Page numbers are likely to change later down the production process.) Please also add a statement to the Methods that highlights that you have completed such a checklist, e.g. "This study is reported as per CONSORT guideline (S1 Checklist)”, or similar.

FIGURES AND TABLES

SUPPLEMENTARY MATERIAL

REFERENCES

RCTs 

* PLOS Medicine requires that all trials be prospectively registered in one of registries recognized by WHO. Please ensure that study registration details are included in the Methods section.

* Please structure the Methods section using the following sub-headings: Study design and participants, Randomization and masking, Procedures, Outcomes, Statistical analysis.

* Please specify the dates (Month Day, Year) during which study enrollment and follow up occurred.

* Please include absolute numbers wherever you report percentages; eg, n/N (%)

* Please present the safety data for the study including numbers of specific events and whether or not adverse events are thought to be related to treatment. AEs should be reported in the abstract, per CONSORT and CONSORT-Harms.

* If your trial had to undergo important modifications in response to extenuating circumstances, please complete the CONSERVE-CONSORT checklist and provide in your Supporting Information; (https://www.equator-network.org/reporting-guidelines/guidelines-for-reporting-trial-protocols-and-completed-trials-modified-due-to-the-covid-19-pandemic-and-other-extenuating-circumstances-the-conserve-2021-statement/). When completing the checklist, please use section and paragraph numbers, rather than page numbers.

* In keeping with our commitment to Open Science, please include the study protocol document and analysis plan (including any amendments) as Supporting Information to be published with the manuscript if accepted.

---

## [Decision Letter · Decision Letter 2]

13 Dec 2024

Dear Dr. Quattrochi,

Thank you very much for re-submitting your manuscript "Effects of a community-driven water, sanitation, and hygiene intervention on diarrhea, child growth, and local institutions: a cluster-randomized controlled trial in rural Democratic Republic of Congo" (PMEDICINE-D-24-02389R2) for review by PLOS Medicine.

I have discussed the paper with my colleagues and the academic editor and it was also seen again by three reviewers. I am pleased to say that provided the remaining editorial and production issues are dealt with we are planning to accept the paper for publication in the journal.

[LINK]

We look forward to receiving the revised manuscript by Dec 20 2024 11:59PM.   

Sincerely,

Rebecca Kirk

On behalf of:

Syba Sunny, MBBS, MRes, FRCPath

Senior Editor 

PLOS Medicine

plosmedicine.org

Requests from Editors:

GENERAL EDITORIAL REQEUSTS

* Please confirm that your title complies with to PLOS Medicine's style. Your title must be nondeclarative and not a question. It should begin with main concept if possible. "Effect of" should be used only if causality can be inferred, i.e., for an RCT. Please place the study design ("A randomized controlled trial," "A retrospective study," "A modelling study," etc.) in the subtitle (ie, after a colon).

* Please confirm that your abstract to complies with our requirements, including providing all the information relevant to this study type https://journals.plos.org/plosmedicine/s/submission-guidelines#loc-abstract

* Please ensure that the Introduction ends with a clear description of the study question or hypothesis.

* Please ensure that all abbreviations are defined at first use throughout the text.

FUNDING STATEMENT

* The funding statement should include: specific grant numbers, initials of authors who received each award, URLs to sponsors’ websites. Also, please state whether any sponsors or funders (other than the named authors) played any role in study design, data collection and analysis, the decision to publish, or preparation of the manuscript. If they had no role in the research, include this sentence: “The funders had no role in study design, data collection and analysis, decision to publish, or preparation of the manuscript.”

COMPETING INTERESTS STATEMENT

* All authors must declare their relevant competing interests per the PLOS policy, which can be seen here: https://journals.plos.org/plosmedicine/s/competing-interests For authors with ties to industry, please indicate whether any of the interests has a financial stake in the results of the current study.

DATA AVAILABILITY

* Thank you for agreeing to make your data available. At this time, please provide the link to the data repository and accession numbers required for access.

FIGURES

* Where data points are discrete, please ensure that they are depicted in the figures as discrete data and not as a continuous line.

Comments from Reviewers:

Reviewer #1: The authors have sufficiently responded to my comments, though the resubmission file did not include the Tables so I was unable to review changes to those. Overall, I still don't feel the manuscript is sufficiently novel to warrant publication in PLOS Medicine given other similar research in this area.

Reviewer #2: General comments

The authors have done a very good job to improve the reporting of this cluster randomized trial. Despite a slightly messy pre-registration and randomization process (at least by biomedical clinical trial standards, which are relevant here), the overall inference from the trial should be reasonable, and the transparent reporting in the revised paper helps clarify what was done and why some of the design characteristics are slightly unintuitive. The authors should be commended on fielding a trial under such challenging circumstances, and for doing a very nice, comprehensive job to respond to referee comments on the previous draft. I had just two small comments on the revised draft. I hope these comments are helpful

Ben Arnold, UCSF

MAJOR comments

(1) Line 195

"In Kasai Central, due to budget constraints, we increased the probability of being selected into the control group proportionally to reflect the fact that only 16 out of 81 villages could receive the intervention."

Please specify the allocation probabilities for the intervention and control groups in Kasai Central. 

MINOR comments

(2) I may have missed it in the SI, but in the results it would be helpful for the authors to report the estimated ICC for diarrhea and LAZ, their primary health outcomes. With much higher precision than planned for in the design, it seems like the ICC for diarrhea might have been lower than the one they assumed in the design (ICC=0.09), and reporting such information is helpful for planning of future studies. From Murray 2004 (an old paper, but important paper https://pmc.ncbi.nlm.nih.gov/articles/PMC1448268/): "Because the ICC of concern in any GRT is the ICC as it operates in the primary analysis, these findings reinforce the need for investigators to use estimates in their power analyses that closely reflect the endpoints, target population, and primary analysis planned for the trial. And while the sources just cited will help considerably in this regard, we join others who have urged publication of such estimates as a routine part of reporting the results of GRTs." 

Reviewer #4: The authors have provided detailed responses to the questions and comments and have incorporated them thoroughly into their manuscript. In my opinion, there are no further objections that would prevent publication.

[LINK]

---

## [Editor Report · Decision Letter 3]

8 Jan 2025

Dear Dr Quattrochi, 

On behalf of my colleagues and the Academic Editor, Zulfiqar Bhutta, I am pleased to inform you that we have agreed to publish your manuscript "Effects of a community-driven water, sanitation, and hygiene intervention on diarrhea, child growth, and local institutions: a cluster-randomized controlled trial in rural Democratic Republic of Congo" (PMEDICINE-D-24-02389R3) in PLOS Medicine.

PRESS

Sincerely, 

Rebecca Kirk

On behalf of:

Syba Sunny, MBBS, MRes, FRCPath 

Senior Editor 

PLOS Medicine